# Dietary *Lactobacillus delbrueckii* Affects Ileal Bacterial Composition and Circadian Rhythms in Pigs

**DOI:** 10.3390/ani14030412

**Published:** 2024-01-26

**Authors:** Wenxin Luo, Zhangzheng Yin, Mingliang Zhang, Xingguo Huang, Jie Yin

**Affiliations:** 1College of Animal Science and Technology, Hunan Agricultural University, Changsha 410128, China; lwxclara0105@163.com (W.L.); yinzhangzheng2021@163.com (Z.Y.); mlzhang111@163.com (M.Z.); hxg68989@hunau.edu.cn (X.H.); 2Hunan Biological and Electromechanical Polytechnic, Changsha 410125, China

**Keywords:** *L. delbrueckii*, ileal bacteria, circadian rhythms, pigs

## Abstract

**Simple Summary:**

Dietary *L. delbrueckii* affected ileal bacterial composition at the genus level, including *Lactobacillus*, *Enterococcus*, *Leptotrichia*, *Pediococcus*, *Bifidobacte*, *Cel-lulosilyticum*, *Desulfomicrobium*, *Sharpea*, *Eubacterium*, *Propionivibrio*, and *Aerococcus*, which were associated with membrane transport, metabolism of cofactors and vitamins, cell motility, the endocrine system, signaling molecules and interaction, and the nervous system. Bacteria were sequenced at 6 Zeitgeber times (ZT), and the results showed that *Lactobacillus*, *Terrisporobacter*, and *Weissella* exhibited significant rhythmic fluctuation in the control pigs, which was disturbed by probiotic exposure. In addition, dietary *L. delbrueckii* shaped circadian rhythms in ileal *Romboutsia*, *Erysipelatoclostridium*, *Celllosilyticum*, and *Eubacterium* abundances. Dietary *L. delbrueckii* affected both ileal bacterial composition and circadian rhythms in pigs.

**Abstract:**

Intestinal bacteria, synchronized with diet and feeding time, exhibit circadian rhythms and anticipate host gut function; however the effect of dietary probiotics on gut bacterial diurnal rhythms remains obscure. In this study, bacteria were sequenced at 6 Zeitgeber times (ZT) from a pig model of ileal T-shaped fistula to test ileal bacterial composition and circadian rhythms after *Lactobacillus delbrueckii* administration. The results showed that dietary *L. delbrueckii* enhanced ileal bacterial α-diversity at Zeitgeber time (ZT) 16, evidenced by an increased Simpson index compared with control pigs. At the phylum level, Firmicutes was identified as the largest phyla represented in pigs, but dietary *L. delbrueckii* only increased the abundance of Tenericutes at ZT16. At the genus level, 11/100 genera (i.e., *Lactobacillus*, *Enterococcus*, *Leptotrichia*, *Pediococcus*, *Bifidobacte*, *Cellulosilyticum*, *Desulfomicrobium*, *Sharpea*, *Eubacterium*, *Propionivibrio*, and *Aerococcus*) were markedly differentiated in *L. delbrueckii*-fed pigs and the effect was rhythmicity-dependent. Meanwhile, dietary *L. delbrueckii* affected six pathways of bacterial functions, such as membrane transport, metabolism of cofactors and vitamins, cell motility, the endocrine system, signaling molecules and interaction, and the nervous system. Cosinor analysis was conducted to test bacterial circadian rhythm in pigs, while no significant circadian rhythm in bacterial α-diversity and phyla composition was observed. *Lactobacillus*, *Terrisporobacter*, and *Weissella* exhibited significant rhythmic fluctuation in the control pigs, which was disturbed by probiotic exposure. In addition, dietary *L. delbrueckii* affected circadian rhythms in ileal *Romboutsia*, *Erysipelatoclostridium*, *Cellulosilyticum*, and *Eubacterium* abundances. Dietary *L. delbrueckii* affected both ileal bacterial composition and circadian rhythms, which might further regulate gut function and host metabolism in pigs.

## 1. Introduction

The animal gut is a complex ecosystem of host cells, microbiota, and available nutrients, and the role of intestinal bacterial balance has been widely addressed in host nutrition metabolism, gut physiology, and immunity [1,2,3,4]. Microbial disturbances are highly associated with gut diseases, such as post-weaning diarrhea in piglets [5,6]. Indeed, abrupt changes in diet and environment are frequently associated with gut microbiota dysbiosis (decreased α-diversity and *Lactobacillus* abundance), which is emerging as a leading cause of gut diseases and metabolic disorders [7,8]. Thus, probiotic species are generally introduced to improve gut health and growth via the modulation of gut microbiota composition in animal production [9]. Currently, most probiotic products are developed with genera *Lactobacilli*, *Bifidobacteria*, *Lactococci*, *Bacillus*, Propionibacterium, and *Saccharomyces*. Several species and strains of *Lactobacilli*, including *Lactobacillus delbrueckii*, *Lactobacillus acidophilus*, *Lactobacillus plantarum*, *Lactobacillus johnsonii*, and *Lactobacillus reuteri*, have been widely studied in the prevention of gut diseases, and there has been a marked improvement in the population of gut microbiota in the pig industry [9,10].

Previous studies have shown that the gut microbiota exhibits daily variations and has the same time period as human or animal central circadian clocks, of approximately 24 h [11,12,13]. We identified 29 similar correlations between gut 29 genera (58% of top 50) and clock genes in the liver and found that bacterial circadian rhythm serves as a key regulator in host metabolism [14]. Indeed, disruption of the microbial circadian system can perturb host metabolism, energy homeostasis, and inflammatory pathways [15]. For example, Brooks et al. reported that intestinal daily rhythms of segmented filamentous bacteria in epithelial attachment drives caused diurnal rhythms in innate immunity against *Salmonella Typhimurium* infection to vary across the day–night cycle [11]. These microbial diurnal rhythms synchronize with feeding rhythms to anticipate host gut function and can be affected by dietary nutrients [14,16]. Circadian transcription factor NFIL3, histone deacetylase, gut-microbe-generated short-chain fatty acids, and rhythmic hormone melatonin were recruited rhythmically, and produced synchronized diurnal oscillations in histone acetylation, metabolic gene expression, and nutrient uptake [17,18,19,20]. Choi et al. systemically reviewed microbial oscillators that are generally promoted by dietary nutrients, such as plant-based, low-fat (lean) diets, and most were abolished by low-fiber, high-sugar, high-fat (Western) diets [16]. However, the response of gut microbial diurnal rhythms and related host metabolism to dietary probiotics has not been fully resolved.

*Lactobacillus delbrueckii* belongs to the *Lactobacillus* genus; it has been demonstrated to exert versatile beneficial effects on modulating intestinal immunity, increasing gut microbial diversity, promoting growth performance, and even preventing disease onset in pigs [21,22,23,24,25]. In this study, we used a pig model of a T-shaped catheter installed in the terminal ileum to investigate the effects of dietary *L. delbrueckii* on bacterial circadian rhythms and metabolism.

## 2. Materials and Methods

### 2.1. Animals and Experimental Design

All protocols and procedures involved in the experiment were approved by the Animal Ethics Committee of Hunan Agricultural University (Changsha, China). Ten healthy Landrace × Yorkshire crossbred growing pigs were installed with a silicone-coated latex T-shaped catheter in the terminal ileum. Briefly, pigs were anesthetized (Zoletil 50, Virbac Co., Nice, France) and then operated on to install the catheter in the terminal ileum. After surgery, all pigs received one week of antibiotic anti-inflammatory treatment and three weeks recovery; they were then randomly divided into two groups (body weight 38.70 ± 5.33 kg, *n* = 5): a control group and an *L. delbrueckii*-supplemented group in which pigs were fed a diet containing 0.1% *L. delbrueckii* (5 × 10^10^ CFU/g) provided by PERFLY-BIO (Changsha, China). The formulation of basal diet contained 66.76% corn, 4% wheat middling, 6% wheat bran, 18% soybean meal (43% crude protein), 1% soybean oil, 0.24% L-lysine, and 4% premix with trace elements and vitamins. The nutrition level included 3414 kcal/kg digestible energy, 14.82% crude protein, 0.85% standardized ileal digestible lysine, 0.6% calcium, and 0.55% phosphorus [22]. Animals were fed twice a day (8:00 a.m. and 15:00 p.m.) and had free access to water. On day 28, ileal chyme was collected from the T-shaped catheter at Zeitgeber times (ZT) 0 (8:00 a.m.), 4 (12:00 a.m.), 8 (16:00 p.m.), 12 (20:00 p.m.), 16 (24:00 p.m.), and 20 (4:00 a.m.) for bacterial sequencing.

### 2.2. Bacterial Sequencing

Total DNA from 60 ileal chyme samples was isolated using a DNeasy PowerSoil kit (Qiagen, Hilden, Germany); 1 ng/uL DNA samples were amplified for 16S rDNA genes of distinct regions (16S V3-V4) using specific primers (515F-806R) with barcodes. Sequencing was performed with an Illumina MiSeq with two paired-end read cycles of 300 bases each (Novogene, Beijing, China). Sequencing libraries were generated using TruSeq^®^ DNA PCR-Free Sample Preparation Kit (Illumina, San Diego, CA, USA) following the manufacturer’s recommendations, and index codes were added. The library quality was assessed on the Qubit@ 2.0 Fluorometer (Thermo Scientific, Waltham, MA, USA) and Agilent Bioanalyzer 2100 systems. Paired-end reads were assigned to samples based on their unique barcode and truncated by cutting off the barcode and primer sequences. Clean reads were subjected to primer sequence removal and clustering to generate operational taxonomic units (OTUs) using VSEARCH software with a 97% similarity cutoff. The representative read of each OTU was selected using the QIIME package. OTU abundance information was normalized using a standard of sequence number corresponding to the sample with the least sequences. All representative reads were annotated and blasted against the Unite database using BLAST. Subsequent analyses of alpha diversity and bacterial composition were all performed based on this output normalized data. Alpha diversity was applied in analyzing complexity of species diversity for a sample through 5 indexes, including observed species, Shannon, Simpson, Chao1, and ACE. Bacterial composition at the phylum and genus levels were compared between control and *L. delbrueckii* groups. Tax4Fun was further used for genome prediction of bacterial function. Briefly, the metagenome was predicted by looking up the precalculated genome content for each OTU, multiplying the normalized OTU abundance by each KEGG abundance in the genome, and summing these KEGG abundances together per sample. The prediction yielded a table of KEGG abundances for each metagenome sample in the OTU table. For optional organism-specific predictions, the per-organism abundances are retained and annotated for each KEGG. In this study, 43 KEEG pathway annotations in KEEG level 2 were identified in the Greengenes reference tree.

### 2.3. Statistical Analysis

All values are given as means ± SEM. Differences between the control and *L. delbrueckii* groups at each ZT point and the average value in 24 h rhythmic cycle were compared using Student’s *t* test (SPSS version 22 software), and variations within groups at different ZT points were tested with Duncan’s multiple comparisons. The rhythmicity of ileal bacteria was assessed with cosinor analysis using a nonlinear regression model within Sigmaplot V10.0 (Systat Software, San Jose, CA, USA) [14]. A *p* value of < 0.05 was considered significant rhythm. The model can be written according to the equation f(x) = A + B cos [2π(x + C)/24], with f(x) indicating relative levels of bacteria, x indicating the time of sampling (h), A indicating the mean value of the cosine curve (midline estimating statistic of rhythm [mesor]), B indicating the amplitude of the curve (half of the sinusoid), and C indicating the acrophase (h).

## 3. Results

### 3.1. Effects of Dietary L. delbrueckii on Ileal Bacterial α-Diversity

Observed species, Shannon, Simpson, Chao1, and ACE indexes were analyzed to evaluate the role of dietary *L. delbrueckii* in ileal bacterial diversity (Figure 1). At ZT8-20, results showed increased bacterial α-diversity; however, the only significant difference was noticed at ZT16, when *L. delbrueckii* enhanced the Simpson index compared with the control pigs (*p* < 0.05). Although the observed species, Shannon, Chao1, and ACE indexes were lowered in the *L. delbrueckii*-fed pigs, the difference was insignificant (*p* > 0.05). We further calculated the average value of bacterial α-diversity in a 24 h rhythmic cycle, with also no significant difference observed. Variations within groups at different ZT points were tested with Duncan’s multiple comparisons and the results concluded that ileal bacterial α-diversity at ZT20 was higher than that at other ZT times, both in control and *L. delbrueckii*-fed pigs.

### 3.2. Effects of Dietary L. delbrueckii on Ileal Bacterial Composition

At the phylum level (Figure 2), Firmicutes was identified as the largest phylum represented in each dataset. Although no difference was observed during different rhythmic times (*p* > 0.05), the average Firmicutes abundance was markedly lower in *L. delbrueckii*-fed pigs (*p* > 0.01). Interestingly, dietary *L. delbrueckii* markedly increased the abundance of Tenericutes at ZT16, but not at other ZT points, and the average level compared with the control group (*p* < 0.05). We failed to notice any significant difference in the abundances of Proteobacteria, Bacteroidetes, Fusobacteria, Cyanobacteria, and Actinobacteria of the *L. delbrueckii*-fed pigs at each ZT time (*p* > 0.05), while the 24 h average abundances of Proteobacteria and Fusobacteria were decreased in response to *L. delbrueckii* exposure (*p* < 0.05).

At the genus level, the top 100 genera were analyzed and 11 bacteria were differentiated in *L. delbrueckii*-fed pigs, including *Lactobacillus*, *Enterococcus*, *Leptotrichia*, *Pediococcus*, *Bifidobacte*, *Cellulosilyticum*, *Desulfomicrobium*, *Sharpea*, *Eubacterium*, *Propionivibrio*, and *Aerococcus* (Figure 3). Although *L. delbrueckii* belongs to the *Lactobacillus* genus, *L. delbrueckii*-fed pigs showed lower abundance of *Lactobacillus* compared with the control pigs, with significance at ZT16 and 20 (*p* < 0.05), indicating that *L. delbrueckii* might inhibit the colonization of other *Lactobacillus*. Similarly, *Bifidobacterium* was reduced at ZT0 in response to *L. delbrueckii* exposure (*p* < 0.05). From ZT4 to 16, *Enterococcus* was markedly increased in *L. delbrueckii*-fed pigs (*p* < 0.05). In addition, dietary supplementation with *L. delbrueckii* markedly increased ileal *Leptotrichia* abundance at ZT4, and *Pediococcus*, *Desulfomicrobium*, *Sharpea*, *Eubacterium*, *Propionivibrio*, and *Aerococcus* levels at ZT16 (*p* < 0.05). However, we found an inconsistency in *Cellulosilyticum* composition, which was increased at ZT4 but decreased at ZT16 in *L. delbrueckii*-fed pigs. For the average level of bacteria in the 24 h rhythmic cycle, ileal *Lactobacillus* and *Bifidobacterium* were decreased and *Enterococcus* was enhanced in *L. delbrueckii*-fed pigs (*p* < 0.05).

### 3.3. Effects of Dietary L. delbrueckii on Ileal Bacterial Functions

Tax4Fun was further used to predict the functional capabilities of bacterial communities based on 16S rDNA datasets [26]; 43 pathways were analyzed and 6 were differentiated in *L. delbrueckii*-fed pigs (Table 1). Dietary *L. delbrueckii* reduced bacterial function related to membrane transport at ZT8, while significantly enhancing metabolism of cofactors and vitamins (ZT16), cell motility (ZT16), the endocrine system (ZT16), signaling molecules and interaction (ZT8), and the nervous system (ZT16) compared with the control pigs (*p* < 0.05). Together, these results indicated that dietary *L. delbrueckii* affected host metabolism via altering gut microbiota composition. For the 24 h average level of bacterial functions, dietary *L. delbrueckii* decreased membrane transport and enhanced metabolism of cofactors and vitamins, cell motility, and endocrine system-related bacterial functions (*p* < 0.05).

### 3.4. Effects of Dietary L. delbrueckii on Ileal Bacterial Circadian Rhythm

Ileal bacteria were sequenced at ZT 0, 4, 8, 12, 16, and 20 and cosinor analysis was conducted to determine the bacterial circadian rhythm in the pigs. All pigs failed to show any significant circadian rhythm in bacterial α-diversity (observed species, Shannon, Simpson, Chao1, and ACE indexes) and phyla composition (Firmicutes, Tenericutes, Proteobacteria, Bacteroidetes, Fusobacteria, Cyanobacteria, and Actinobacteria) (*p* > 0.05). At the genus level, 7/100 genera were identified to be circadian-rhythmic-changed in control and *L. delbrueckii*-fed pigs (Table 2), including *Lactobacillus*, *Romboutsia*, *Terrisporobacter*, *Erysipelatoclostridium*, *Cellulosilyticum*, *Eubacterium*, and *Weissella*. Among them, *Lactobacillus*, *Terrisporobacter*, and *Weissella* exhibited a significant rhythmic fluctuation in the control pigs (*p* < 0.05), while the circadian clock disappeared in the *L. delbrueckii*-fed pigs, indicating the destructive effect of *L. delbrueckii* on ileal *Lactobacillus*, *Terrisporobacter*, and *Weissella* abundances. Meanwhile, there was no significant circadian rhythm in ileal *Romboutsia*, *Erysipelatoclostridium*, *Cellulosilyticum*, and *Eubacterium* abundances, which was reprogrammed in *L. delbrueckii*-fed pigs (*p* < 0.05).

The circadian rhythm of bacterial functions was also analyzed and 13 pathways were markedly rhythmically altered in control and *L. delbrueckii*-fed pigs (Table 3). Bacterial functions related to enzyme families and metabolism of terpenoids and polyketides showed significant circadian rhythms both in control and *L. delbrueckii*-fed pigs. Nine bacterial pathways markedly fluctuated in control pigs but not in *L. delbrueckii*-fed pigs, including lipid metabolism, folding, sorting and degradation, transport and catabolism, cellular community prokaryotes, xenobiotics biodegradation and metabolism, endocrine and metabolic diseases, the viral protein family, neurodegenerative diseases, and the digestive system. We also observed a significant circadian clock of cellular processes and signaling and signaling molecules and interaction in *L. delbrueckii*-fed pigs but not in the control animals. In summary, the circadian rhythms of bacterial composition and functions were confirmed in pigs, and the clock could be affected by dietary *L. delbrueckii*.

## 4. Discussion

Use of lactobacilli as probiotics in the pig industry has been gaining attention due to their ability to improve growth performance, host immune systems, carcass quality, gut metabolic capacities, and balance in gut microbiota composition [27]. We have previously shown that dietary supplementation with *L. delbrueckii* during the suckling period showed long-lasting immunomodulatory impacts on intestinal immunity after weaning via accelerating dendritic cell maturation and activation [25]. In weaning or lipopolysaccharide-challenged piglets, supplementation of LAB improved intestinal immune response (increasing secretory immunoglobulin A) and antioxidant capacity (decreasing serum and hepatic malondialdehyde and enhanced the activity of hepatic glutathione peroxidase) [28]. The growing-finishing pigs also experienced an improvement in gut function and metabolism after *L. delbrueckii* administration.

The direct target of dietary *L. delbrueckii* in pigs is the gut microbiota. However, we failed to notice any significant difference in bacterial α-diversity, which was similar to our previous study, which found that dietary probiotics (such as *Lactobacillus johnsonii*) did not affect microbial diversity [1]. In addition, lower abundances of *Lactobacillus and Bifidobacterium* were observed in *L. delbrueckii*-fed pigs, the reason possibly associated with a promoting effect on other probiotic species, such as *Enterococcus* and *Pediococcus*. Similarly, dietary fermented milk of cheese-derived *L. delbrueckii* increased gut *Enterococcus sp.* abundance in the murine model with alcohol-induced hepatic injury [29]. However, all these effects were rhythmicity-dependent. For example, the Cellulosilyticum composition increased at ZT4 but decreased at ZT16 in *L. delbrueckii*-fed pigs compared with the control pigs. In addition, most microbiota differentiated at ZT16, the reason possibly associated with the feeding rhythm, with pigs fed at ZT0 and ZT7 showing lower nutrient contents in the ileum [30,31]. Meanwhile, the highly secreted pineal melatonin under low ambient light levels at night exhibited a close association with the gut microbial alterations and rhythms [32,33]. Indeed, we previously found a marked effect of melatonin on gut *Bacteroides*, *Alistipes*, *Parasutterella*, and *Lactobacillus* abundances [34]. Further studies are needed to fully study the effects and mechanisms of dietary probiotics on gut microbiota composition.

The gut microbiota performs some basic functions in the immunological, metabolic, structural, and neurological landscapes of animals, and recent developments in genome sequencing technologies have enabled researchers to explore the microbiota and in particular its functions at a more detailed level. Tax4Fun, an open-source R package, has been reported to predict the microbial functional profiles obtained from metagenomic shotgun sequencing approaches [26]. Here, metabolism of cofactors and vitamins, cell motility, the endocrine system, signaling molecules and interaction, and the nervous system were enhanced in *L. delbrueckii*-fed pigs. Santos et al. also characterized a vitamin-specific ECF-type ABC transporter from *L. delbrueckii* [35], indicating a role in vitamin metabolism. Indeed, genomic characterization of *L. delbrueckii* strains uncovered a high proteolytic and metabolic activity, able to produce B-complex vitamins [36]. Comparative genomics analyses also revealed that different *L. delbrueckii* probiotic strains shared essential genes related to acid and bile stress response, antimicrobial activity, and inflammatory signaling pathways (TLR2/4-MAPK, TLR2/4-NF-κB, and NOD-like receptors) [36]. In addition, the potential mechanisms of the gut microbiota in host metabolism might be associated with bile acid circulation, short-chain fatty acids, bacteriocins, miRNA, or bacterial receptors [37]. We also noticed an improvement of bile acid enterohepatic circulation and lipid metabolism in *L. delbrueckii*-fed pigs [22]. However, the detailed mechanism of how dietary *L. delbrueckii* affects host metabolism needs further study.

Another important finding from the present study is the circadian rhythms of bacterial composition and functions in pigs. Here, a T-shaped catheter installed in the terminal ileum was first used to investigate the bacterial circadian rhythms in pigs, while bacterial diversity and composition at the phylum level were not rhythmically changed, which was inconsistent in murine studies where Firmicutes and Bacteroidetes exhibited significant rhythmicity [14]. At the genus level, most microbiota exhibited a marked daily rhythmicity, including *Lactobacillus*, *Anaerotruncus*, *Parasutterella*, *Alloprevotella*, *Parabacteroides*, *Alistipes*, *Oscillibacter*, *Rikenella*, *Lachnoclostridium*, and *Ruminiclostridium* [14]. However, only 7/100 genera were identified to be circadian-rhythmic-changed in pigs; this might be associated with animal model and drug use, because antibiotics were used after the ileal T-shaped fistula surgery. Antibiotic treatment has been reported to underscore the essential role of gut microbes in clock reprogramming and hepatic circadian homeostasis, and the mechanism might be caused by the gut microbiota involved in PPARγ-mediated activation of newly oscillatory transcriptional programs [38]. Meanwhile, current data indicate that circadian rhythms in bacterial composition and functions can be disturbed by dietary probiotics. Indeed, the role of *Lactobacillus* has been confirmed in promoting anti-inflammatory responses, maintaining intestinal epithelial homeostasis, and modulating the circadian–metabolic axis in its host [39]. In addition, the circadian rhythm of the gut microbiota can also be affected by host rhythmic signals, such as melatonin and other metabolites secreted into gut lumen. We previously found that melatonin treatment reprogramed gut microbial composition in diet-induced obesity [34]. Although the current data indicate a potential role for dietary *L. delbrueckii* in gut microbial composition and circadian rhythms, the conclusion was limited because of animal number, sampling times, feeding times, and phenotypic and physiological metabolic analysis. Thus, further studies with more pigs, a 48 h rhythmic period, metabolic potential, and other stains in different animal models are suggested to confirm the merit of dietary probiotics in animal production.

## 5. Conclusions

Dietary *L. delbrueckii* affected ileal bacterial composition at the genus level, including *Lactobacillus*, *Enterococcus*, *Leptotrichia*, *Pediococcus*, *Bifidobacte*, *Cellulosilyticum*, *Desulfomicrobium*, *Sharpea*, *Eubacterium*, *Propionivibrio*, and *Aerococcus*, which were associated with membrane transport, metabolism of cofactors and vitamins, cell motility, the endocrine system, signaling molecules and interaction, and the nervous system. Bacteria were sequenced at 6 ZT, and the results showed that *Lactobacillus*, *Terrisporobacter*, and *Weissella* exhibited significant rhythmic fluctuation in the control pigs, which was disturbed by probiotic exposure. In addition, dietary *L. delbrueckii* shaped circadian rhythms in ileal *Romboutsia*, *Erysipelatoclostridium*, *Celllosilyticum*, and *Eubacterium* abundances. In conclusion, dietary *L. delbrueckii* affected ileal bacterial composition and circadian rhythms in pigs, which also indicated a potential implication for gut microbial manipulation to improve animal (also including poultry and ruminants) production, and even for human health.

## Figures and Tables

**Figure 1 animals-14-00412-f001:**
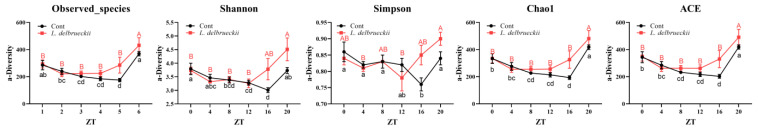
Effects of dietary *L. delbrueckii* on ileal bacterial α-diversity. a–d (control group) or A,B (*L. delbrueckii* group) within a row means different superscripts differ between other ZT points using the Duncan’s multiple comparisons test (*p* < 0.05). The same as below.

**Figure 2 animals-14-00412-f002:**
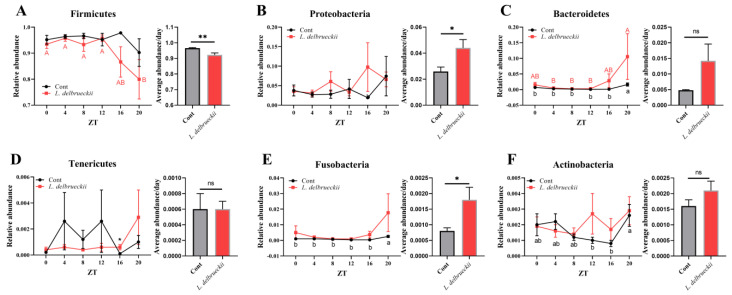
Effects of dietary *L. delbrueckii* on ileal bacterial phyla. (**A**) Firmicutes abundance; (**B**) Proteobacteria abundance; (**C**) Bacteroidetes abundance; (**D**) Tenericutes abundance; (**E**) Fusobacteria abundance; (**F**) Actinobacteria abundance. a,b (control group) or A,B (*L. delbrueckii* group) within a row means different superscripts differ between other ZT points using the Duncan’s multiple comparisons test (*p* < 0.05). * means the difference was significant compared with the control (cont) group using Student’s *t* test (*p* < 0.05). ** means the difference was significant compared with the control (cont) group using Student’s *t* test (*p* < 0.01). ns means the difference was insignificant (*p* > 0.05).

**Figure 3 animals-14-00412-f003:**
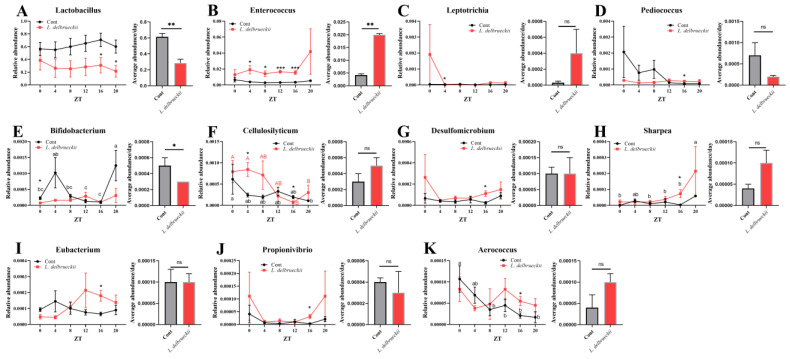
Effects of dietary *L. delbrueckii* on ileal bacterial genera. (**A**) *Lactobacillus* abundance; (**B**) Enterococcus abundance; (**C**) Leptotrichia abundance; (**D**) Pediococcus abundance; (**E**) Bifidobacterium abundance; (**F**) Cellulosilyticum abundance; (**G**) Desulfomicrobium abundance; (**H**) Sharpea abundance; (**I**) Eubacterium abundance; (**J**) Propionivibrio abundance; (**K**) Aerococcus abundance. a–c (control group) or A-B (*L. delbrueckii* group) within a row means different superscripts differ between other ZT points using the Duncan’s multiple comparisons test (*p* < 0.05). * means the difference was significant compared with the control (cont) group using Student’s *t* test (*p* < 0.05). ** means the difference was significant compared with the control (cont) group using Student’s *t* test (*p* < 0.01). *** means the difference was significant compared with the control (cont) group using Student’s *t* test (*p* < 0.001). ns means the difference was insignificant (*p* > 0.05).

**Table 1 animals-14-00412-t001:** Effects of dietary *L. delbrueckii* on ileal bacterial functions.

Item	ZT0	ZT4	ZT8	ZT12	ZT16	ZT20	Average
Membrane transport							
Cont	0.1185 ± 0.0011 b	0.1196 ± 0.0006 b	0.1196 ± 0.0003 b	0.1197 ± 0.0007 b	0.1197 ± 0.0006 b	0.1120 ± 0.0016 a	0.1182 ± 0.0006
*L. delbrueckii*	0.1120 ± 0.0046 ab	0.1180 ± 0.0009 a	0.1181 ± 0.0006 a *	0.1181 ± 0.0009 a	0.1169 ± 0.0021 a	0.1045 ± 0.0071 b	0.1146 ± 0.0016 *
Metabolism of cofactors and vitamins							
Cont	0.0299 ± 0.0005 b	0.0303 ± 0.0004 ab	0.0306 ± 0.0003 ab	0.0304 ± 0.0004 ab	0.0298 ± 0.0003 b	0.0313 ± 0.0003 a	0.0304 ± 0.0002
*L. delbrueckii*	0.0310 ± 0.0004	0.0310 ± 0.0003	0.0311 ± 0.0003	0.0309 ± 0.0002	0.0309 ± 0.0003 *	0.0319 ± 0.0004	0.0311 ± 0.0001 **
Cell motility							
Cont	0.0137 ± 0.0018 b	0.0152 ± 0.0015 b	0.0161 ± 0.0007 b	0.0158 ± 0.0013 b	0.0141 ± 0.0011 b	0.0199 ± 0.0004 a	0.0158 ± 0.0006
*L. delbrueckii*	0.0153 ± 0.0014	0.0175 ± 0.0010	0.0183 ± 0.0009	0.0173 ± 0.0008	0.0181 ± 0.0009 *	0.0186 ± 0.0022	0.0175 ± 0.0005 *
Endocrine system							
Cont	0.0067 ± 0.0001 b	0.0066 ± 0.0001 b	0.0067 ± 0.0001 b	0.0067 ± 0.0001 b	0.0065 ± 0.0001 b	0.0071 ± 0.0001 a	0.0067 ± 0.0001
*L. delbrueckii*	0.0071 ± 0.0003 ab	0.0068 ± 0.0001 ab	0.0068 ± 0.0001 ab	0.0068 ± 0.0001 b	0.0069 ± 0.0001 ab *	0.0076 ± 0.0005 a	0.0070 ± 0.0001 *
Signaling molecules and interaction							
Cont	0.0024 ± 0.0001 a	0.0024 ± 0.0001 a	0.0024 ± 0.0000 a	0.0025 ± 0.0001 a	0.0024 ± 0.0001 a	0.0021 ± 0.0001 b	0.0024 ± 0.0000
*L. delbrueckii*	0.0025 ± 0.0000 a	0.0027 ± 0.0001 a	0.0027 ± 0.0001 a *	0.0025 ± 0.0001 a	0.0024 ± 0.0001 ab	0.0021 ± 0.0001 b	0.0025 ± 0.0001
Nervous system							
Cont	0.0018 ± 0.0001 b	0.0018 ± 0.0001 b	0.0018 ± 0.0000 b	0.0018 ± 0.0001 b	0.0017 ± 0.0001 b	0.0022 ± 0.0001 a	0.0018 ± 0.0000
*L. delbrueckii*	0.0020 ± 0.0002 b	0.0019 ± 0.0001 b	0.0019 ± 0.0000 b	0.0019 ± 0.0000 b	0.0020 ± 0.0001 b *	0.0024 ± 0.0002 a	0.0020 ± 0.0001

Note: * means the difference was significant compared with the control (cont) group using Student’s *t* test (*p* < 0.05), ** means the *p* value < 0.01. a,b within a row means different superscripts differ between other ZT points using the Duncan’s multiple comparisons test (*p* < 0.05). The same as below.

**Table 2 animals-14-00412-t002:** Effects of dietary *L. delbrueckii* on ileal bacterial circadian rhythm.

Item	Mesor	Amplitude	Acrophase (h)	*p* Value
*Lactobacillus*				
Cont	0.61	0.07	14.62	0.03
*L. delbrueckii*	0.28	0.02	23.55	0.89
Romboutsia				
Cont	0.04	0.01	5.07	0.06
*L. delbrueckii*	0.08	0.02	8.33	0.05
Terrisporobacter				
Cont	0.02	0.01	2.65	0.01
*L. delbrueckii*	0.03	0.01	3.36	0.26
Erysipelatoclostridium				
Cont	0.00	0.00	18.52	0.46
*L. delbrueckii*	0.00	0.00	16.29	0.00
Cellulosilyticum				
Cont	0.00	0.00	1.61	0.75
*L. delbrueckii*	0.00	0.00	3.58	0.00
Eubacterium				
Cont	0.00	0.00	3.76	0.11
*L. delbrueckii*	0.00	0.00	14.18	0.01
Weissella				
Cont	0.00	0.00	3.70	0.02
*L. delbrueckii*	0.00	0.00	12.18	0.72

Notes: The rhythmicity was assessed with cosinor analysis and a *p* value of < 0.05 was considered significant rhythm. The model of cosinor analysis was according to the equation f(x) = A + B cos [2π(x + C)/24], with f(x) indicating relative levels of bacteria, x indicating the time of sampling (h), A indicating the mean value of the cosine curve (midline estimating statistic of rhythm [mesor]), B indicating the amplitude of the curve (half of the sinusoid), and C indicating the acrophase (h).

**Table 3 animals-14-00412-t003:** Effects of dietary *L. delbrueckii* on the circadian rhythm of bacterial functions prediction.

Item	Mesor	Amplitude	Acrophase (h)	*p* Value
Enzyme families				
Cont	0.03	0.00	8.29	0.04
*L. delbrueckii*	0.03	0.00	6.62	0.02
Lipid metabolism				
Cont	0.03	0.00	8.49	0.03
*L. delbrueckii*	0.03	0.00	7.81	0.06
Folding, sorting and degradation				
Cont	0.02	0.00	20.87	0.01
*L. delbrueckii*	0.02	0.00	20.50	0.09
Transport and catabolism				
Cont	0.02	0.00	20.82	0.04
*L. delbrueckii*	0.02	0.00	20.63	0.09
Cellular community prokaryotes				
Cont	0.02	0.00	8.64	0.01
*L. delbrueckii*	0.02	0.00	8.54	0.10
Cellular processes and signaling				
Cont	0.02	0.00	8.53	0.15
*L. delbrueckii*	0.02	0.00	7.64	0.04
Xenobiotics biodegradation and metabolism				
Cont	0.01	0.00	9.01	0.01
*L. delbrueckii*	0.01	0.00	8.53	0.07
Metabolism of terpenoids and polyketides				
Cont	0.01	0.00	8.65	0.02
*L. delbrueckii*	0.01	0.00	7.48	0.04
Endocrine and metabolic diseases				
Cont	0.00	0.00	20.45	0.04
*L. delbrueckii*	0.00	0.00	19.72	0.09
Signaling molecules and interaction				
Cont	0.00	0.00	8.41	0.25
*L. delbrueckii*	0.00	0.00	6.86	0.03
Viral protein family				
Cont	0.00	0.00	8.36	0.03
*L. delbrueckii*	0.00	0.00	7.74	0.06
Neurodegenerative diseases				
Cont	0.00	0.00	8.53	0.01
*L. delbrueckii*	0.00	0.00	8.11	0.06
Digestive system				
Cont	0.00	0.00	22.27	0.05
*L. delbrueckii*	0.00	0.00	21.63	0.10

Notes: The rhythmicity was assessed with cosinor analysis and a *p* value of < 0.05 was considered significant rhythm. The model of cosinor analysis was according to the equation f(x) = A + B cos [2π(x + C)/24], with f(x) indicating relative levels of bacteria, x indicating the time of sampling (h), A indicating the mean value of the cosine curve (midline estimating statistic of rhythm [mesor]), B indicating the amplitude of the curve (half of the sinusoid), and C indicating the acrophase (h).

## Data Availability

The data presented in this study are available on request from the corresponding author. The availability of the data is restricted to investigators based in academic institutions.

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
