# Peer review of "Dietary Lactobacillus delbrueckii Affects Ileal Bacterial Composition and Circadian Rhythms in Pigs"

_animals, 2024, doi:10.3390/ani14030412_

Round 1

Reviewer 1 Report

Comments and Suggestions for Authors

This manuscript explores the impact of dietary probiotics, specifically Lactobacillus delbrueckii, on the gut microbiota and circadian rhythms in pigs. The authors addressed a significant and under-explored area in probiotics research, focusing on the effects of Lactobacillus delbrueckii on gut microbiota and circadian rhythms. The experimental design involving Landrace×Yorkshire crossbred pigs, the use of a T-shape catheter for sample collection, and detailed sequencing methods indicate a thorough approach to data collection. The research provides valuable insights into the rhythmicity-dependent effects of L. delbrueckii, which could influence future probiotics research and applications in animal health and nutrition.

Despite the thorough methodological approach, the study notes a lack of significant differences in bacterial α-diversity and certain phyla compositions, which might limit the scope of the conclusions drawn.

In particular, as can be seen from Tables 1, 2, 3, and 4, most of the significant differences occurred at the time point ZT16, while there were almost no differences between treatments at other time points, including the later ZT20. Without a reasonable explanation, this would call into question the credibility of the data in this study.

From the results in Table 5, the authors did not compare differences in data between treatments. What the p-values in the table represent is also not stated in the table notes.

The study is specific to a particular probiotic strain and pig breed, which may limit the generalizability of the findings to other strains, species, or broader agricultural contexts. The authors should discuss this.

Line 77-78: Please provide the feed formulation and nutritional levels of the experimental pigs. Material methods are too simple for reproduction in this study.

Line 83: how many samples for 16s sequencing? Please provide the number.

Statistical analysis:

There were six time points in this study, which should have been analyzed using a repeated measures statistical analysis model rather than Student's t test.

Comments on the Quality of English Language

There are several grammatical issues observed throughout the manuscript.

For example:

Abstract:

"Intestinal bacteria are synchronized with diet and time of feeding to exhibit circadian rhythms and anticipate host gut function while the effect of dietary probiotics on gut bacterial diurnal rhythms is obscure."

Change to "Intestinal bacteria, synchronized with diet and feeding time, exhibit circadian rhythms and anticipate host gut function, while the effect of dietary probiotics on gut bacterial diurnal rhythms remains obscure."

Introduction:

"Microbial disturbance are highly associated with gut diseases such as "

should be "Microbial disturbances are highly associated with gut diseases, such as "

Results:

"Although the observed species Shannon Chao1 and ACE were lowered in the L. delbrueckii-fed pigs the difference was insignificant" lacks necessary punctuation and could be improved for clarity.

"while the circadian clock was disappeared in the L. delbrueckii-fed pigs"

Suggested Correction: "while the circadian clock disappeared in the L. delbrueckii-fed pigs"

Discussion:

"The growing-finishing pigs also experimented an improvement of gut function and metabolism after L. delbrueckii administration."

Suggested Correction: "The growing-finishing pigs also experienced an improvement in gut function and metabolism after L. delbrueckii administration."

"For example Cellulosilyticum composition was increased at ZT4 but decreased at ZT16 in L. delbrueckii-fed pigs compared with the control pigs."

Suggested Correction: "For example, the Cellulosilyticum composition increased at ZT4 but decreased at ZT16 in L. delbrueckii-fed pigs compared with the control pigs."

Conclusions:

"Together dietary L. delbrueckii affected ileal bacterial compositions and circadian rhythms in pigs."

Suggested Correction: "In conclusion, dietary L. delbrueckii affects ileal bacterial compositions and circadian rhythms in pigs."

Overall, the manuscript offers valuable insights into the effects of Lactobacillus delbrueckii on ileal bacterial compositions and circadian rhythms in pigs, with a few areas needing improvement in terms of diversity analysis and grammatical precision.

Author Response

Ref: Submission ID animals-2777238

Dear Editors and Reviewers,

First, we would like to thank you and the reviewers for the thoughtful comments and/or suggestions that have helped us to improve the manuscript. Below are the responses to the general questions raised by the reviewers and all revisions have been marked in red in the main text. Notably, we also analyzed the average value in 24-rhythmic cycle and found dietary L. delbrueckii reduced Firmicutes and increased Proteobacteria and Fusobacteria compared with the control pigs. In addition, the variations within groups at different ZT points were tested by the Duncan’s multiple comparisons. And more details can be found at the result section (marked in red color). We hope the revised manuscript is now suitable for publication in the Animals.

Thank you very much for your attention and consideration. We deeply appreciate your consideration of our manuscript. If you have any queries, please don’t hesitate to contact me at the address below.

Sincerely yours,

Jie Yin
[email protected]

Reviewer 2 Report

Comments and Suggestions for Authors

The paper show the effect of Lactobacillus delbrueckii administration on ileal bacterial compositions and circadian rhythms in a pig model of ileal T-shaped fistula. This study is interesting from severalaspect but also requirement some improvement in the current stage.

The introduction part is very small and should be further increased for more background and provide evidence and good references about circadian rhythems. How bacterial fluctuation relates to circadian rhythem should be further clarified though support from review of literature and discussion in your paper. The objectives should also be further clarified and must match with the introduction.

The experimetnl design should be further clarified with detail animals health, the techniques and methodologies applied for bacterial sequencing.

The statistical analysis need further elaboration and discussion.

Implications on GIT function and host metabolism, how it is affected by bacterial α-diversity and composition should be well clarified. Further more, discussion part should include limitations and confounding factors which may affect these parameters and future strategies.

Moreover, circadian rhythms analysis needs further clarification regarding cosinor analysis and more discussion is needed on mechanism how circadian rhythem could affect the current bacterial α-diversity and phyla compositions

Author Response

(The authors gave the same response as above.)

Reviewer 3 Report

Comments and Suggestions for Authors

Dear Authors

The objective of the present work is interesting. However in my opinion the experiment has some dificulties . Feeding system (two times a day might affect the microbial composition. Few numbers of animals etc.

Despite many adverse effects produced by the experimental design it is important to poblish this data.

Author Response

(The authors gave the same response as above.)

Round 2

Reviewer 1 Report

Comments and Suggestions for Authors

After revision, the authors addressed some problems.  But there are still some problems that need to be revised before publication.

1. The paper could benefit from a clearer explanation of the study's relevance to broader applications in animal health and potential implications for human health.

2. The discussion section might be expanded to explore the mechanisms underlying the observed effects more deeply.

3. Some statistical data and results could be better visualized through charts or graphs for easier comprehension. These extensive tables in the manuscript may overwhelm readers.

Author Response

Ref: Submission ID animals-2777238

Dear Editors and Reviewers,

Thank you again for the thoughtful comments and/or suggestions that have helped us to improve the manuscript. Below are the responses to the general questions raised by the reviewers and all revisions have been marked in red in the main text. We hope the revised manuscript is now suitable for publication in the Animals.

Thank you very much for your attention and consideration. We deeply appreciate your consideration of our manuscript. If you have any queries, please don’t hesitate to contact me at the address below.

Sincerely yours,

Jie Yin
[email protected]
